# Critical roles of *Drosophila* ubiquitin carboxyl-terminal hydrolase in eye development

Trang Thi Thuy Cao[1] , Tuan Anh Nguyen[1], Minh Huy Cong Nguyen[1], Trang Thi Huyen Ngo Trang[1] ,
Thao Thi Phuong Dang[1,2,3]

**Ubiquitin C-terminal hydrolase L1 (UCH-L1) is a protein in the ubiquitin–proteasome system involved in numerous cellular processes and implicated in various human diseases, including cancer, neurodegenerative diseases, and metabolic syndrome. However, its roles in development remain poorly understood. In this study, we used *Drosophila melanogaster* as a model to investigate the function of UCH-L1. The *Drosophila* homolog of UCH-L1, *Uch*, was specifically knocked down in the eye imaginal disk using RNA interference (RNAi). The results showed that loss of Uch function induced a rough eye phenotype characterized by abnormal ommatidia, including disorganized arrangement, variable sizes, and irregular orientations. In addition, Uch knockdown significantly affected cone cells, photoreceptor cells, and pigment cells. Remarkably, these defects were fully rescued when *rho-1*, a protease that releases the Spitz, the ligand of EGFR signaling, was co-knocked down. Taken together, these findings strongly demonstrated that Uch plays a crucial role in the development of various *Drosophila* eye cell types and functions in coordination with rhomboid-1 in the *Drosophila* EGFR signaling pathway.**

## Introduction

Ubiquitin C-terminal hydrolase L1 (UCH-L1) is a 25-kD protein that belongs to the family of deubiquitinating enzymes (DUBs) (1, 2). It binds to and stabilizes monoubiquitin in neurons (3). UCH-L1 is predominantly expressed in the brain, accounting for up to 5% of total neuronal protein content (4). It plays a critical role in the ubiquitin–proteasome system, which maintains cellular homeostasis by degrading damaged or misfolded proteins (5). In addition to its role in protein degradation, UCH-L1 interacts with proteins located in various cellular compartments, including the nucleus, cytoplasm, endoplasmic reticulum, plasma membrane, mitochondria, and peroxisomes (5). These widespread interactions

suggest that UCH-L1 regulates multiple cellular processes and is consequently involved in the pathogenesis of various human diseases (6). Previous studies have implicated UCH-L1 in the development and progression of several disorders, including cancer, Parkinson's disease, and type 2 diabetes mellitus (7).

In *Drosophila melanogaster*, the protein homolog of UCH-L1 is *Drosophila* ubiquitin carboxyl-terminal hydrolase (Uch), encoded by the *Uch* gene (CG4265) (8). The identity and similarity between Uch and UCH-L1 were ~44.5% and 75.7%, respectively (8). The Cys90 and the His161 side chains are required for the hydrolase activity of human UCH-L1 and are conserved in *D. melanogaster* (8). In previous studies, the overexpression of the *Uch* gene induced apoptosis, compensatory proliferation, and down-regulation of the MAPK pathway in the fly eye (9).

The adult *D. melanogaster* compound eye is composed of about 700 to 800 ommatidia arranged in an orderly manner (10, 11). Each ommatidium consists of clusters of more than 20 cells, classified into three cell types: photoreceptor cells, cone cells, and pigment cells (10). In addition, the eye structure includes a pseudocone, mechanosensory bristles, and a corneal lens located at the apical surface of the ommatidial unit (12). The *Drosophila* eye originates during the larval stage from a specialized epithelial cell layer known as the eye-antennal imaginal disk. Throughout development, the eye disk undergoes extensive growth and differentiation to form a complete adult eye (10). The differentiation of photoreceptor cells, cone cells, and primary pigment cells is completed at ~20 h after pupal formation (APF) (13). Around 22 h APF, the cells begin to reorganize their structures in a precise and regular pattern. Secondary and tertiary pigment cells then establish a boundary that separates each ommatidium from its neighbors (13, 14). By around 24 h APF, cells that do not directly contact and receive signals from primary pigment cells, and that do not participate in the formation of the pigment cell network are eliminated through apoptosis (13, 15, 16).

The development of the compound eye is tightly regulated and involves several signaling pathways and numerous genes (17). Cells receive signals from neighboring cells that express relevant genes associated with key signaling pathways such as EGFR (epidermal

---

[1]Department of Molecular and Environmental Biotechnology, Faculty of Biology and Biotechnology, University of Science, Ho Chi Minh City, Vietnam   [2]Vietnam National University, Ho Chi Minh City, Vietnam   [3]Laboratory of Molecular and Biotechnology, University of Science, Ho Chi Minh City, Vietnam

Correspondence: thaodp@hcmus.edu.vn

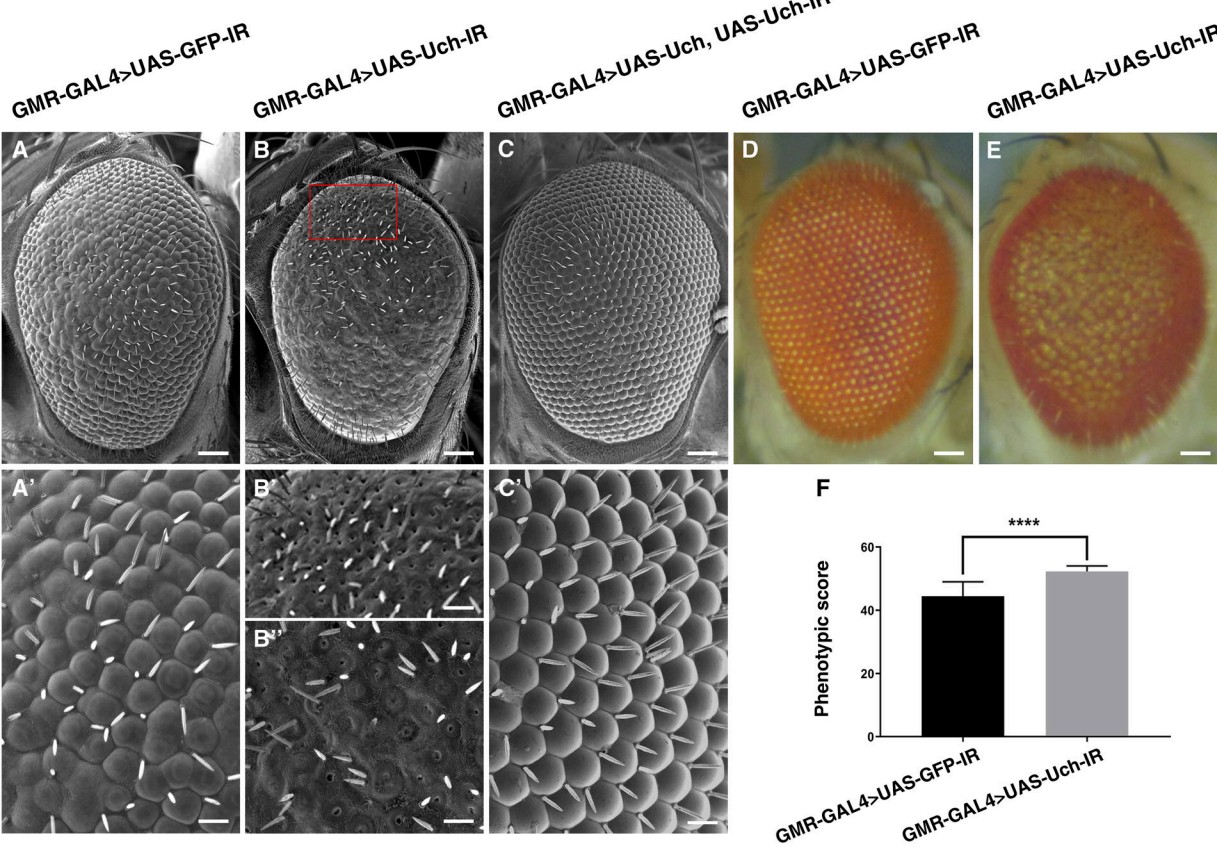

**Figure 1. Knockdown of *Uch* induced a rough eye phenotype.**
**(A, A', D)** Control fly, GMR-GAL4>UAS-GFP-IR. **(B, B', B", E)** Uch knockdown fly, GMR-GAL4>UAS-Uch-IR. **(C)** Fly carrying Uch overexpression and knockdown simultaneously, GMR-GAL4>UAS-Uch, UAS-Uch-IR. **(A, A', B, B', B", C, C', D, E)** Scale bar: 50 μm in (A, B, C, D, E); 14.2 μm in (A', B', B", C'). **(F)** Phenotypic score of the retinae; the error bar represents the SD; statistical analysis was carried out by Welch's *t* test, ****$P$ = 0.0008, n = 6.

growth factor receptor), Notch, Decapentaplegic (BMP), Wingless (Wnt), and Hedgehog (18, 19, 20, 21). Among these, EGFR signaling has been shown to play a crucial role in the differentiation of most eye cells during *Drosophila* eye development (18, 22). Although many previous studies have reported on the regulation of eye development, particularly in *Drosophila*, the roles of Uch in this process remain unclear. In this study, we present several lines of evidence that provide new insight into the essential function of Uch in *Drosophila* eye development.

## Results

### Knockdown of the *Uch* gene induced an abnormal eye phenotype in flies

To investigate the role of Uch in fruit fly eye development, we used the GMR-GAL4 driver to express double-stranded RNA targeting the *Uch* gene specifically posterior to the morphogenetic furrow (MF) in developing fly eye cells. This led to a reduced expression level of the corresponding protein (Fig S1A). Knockdown of *Uch* resulted in a rough eye phenotype

(Fig 1B, B', B", and F) and a loss of pigmentation (Fig 1E) compared with control fly eyes (Fig 1A, A', and D). Interestingly, the rough eye phenotype was rescued by restoring *Uch* expression using a UAS-Uch transgenic line in the Uch loss-of-function background (Fig 1C and C'). Moreover, similar rough eye phenotypes were observed when alternative *Uch* RNAi lines were used (Fig S1B), indicating that the phenotype resulted from specific *Uch* knockdown rather than off-target effects.

From that initial observation, we further examined the external structure of *Uch* down-regulated eyes using the scanning electron microscope. Control eyes expressing GFP dsRNA displayed minor structural defects, including slight bulging, ommatidium fusion, and occasional shallow valleys between central ommatidia, whereas the peripheral regions remained relatively intact (Fig 1A and A'). In contrast, *Uch* knockdown eyes exhibited pronounced morphological abnormalities, characterized by large areas of flattened ommatidia (Fig 1B and B'). These ommatidia either failed to properly develop corneal lenses or underwent extensive fusion resulting in poorly defined boundaries between adjacent ommatidia (Fig 1B"). In addition, contiguous hole-like structures were observed, particularly in the dorsal region of the eye (Fig 1B').

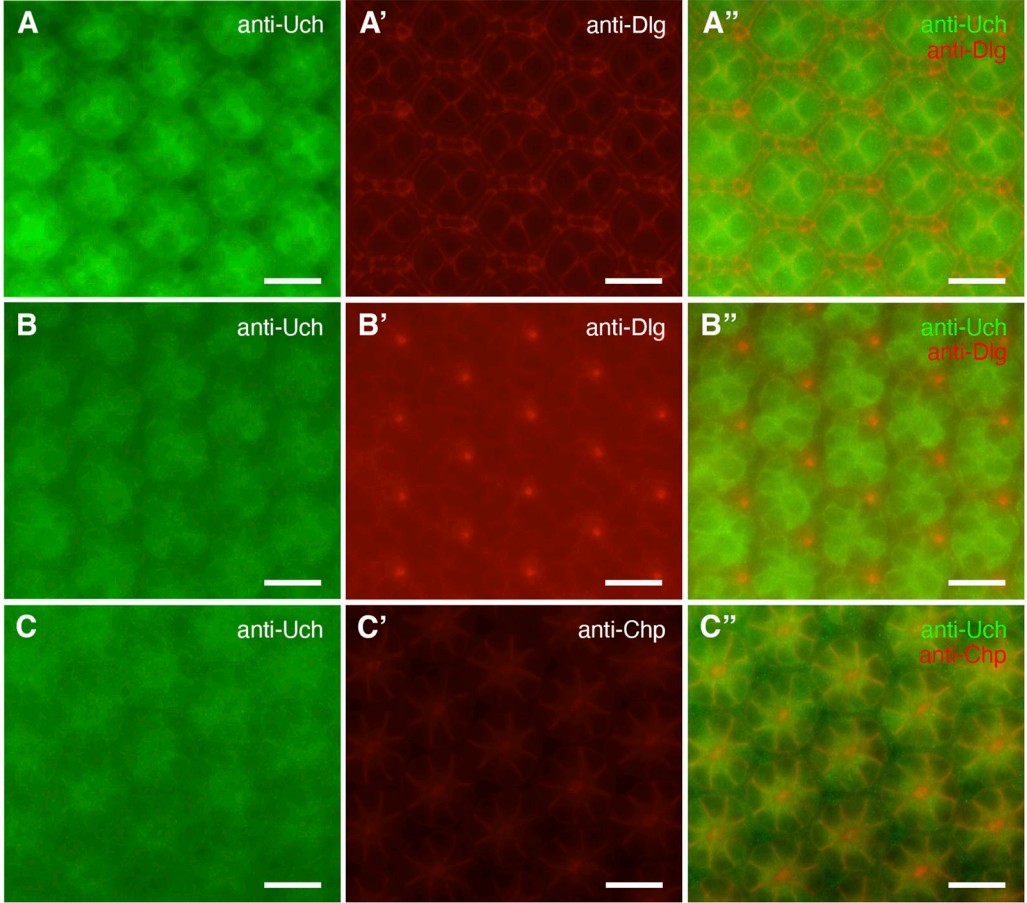

**Figure 2. Location of the protein Uch in the 42-h pupal retina of the fly.**
**(A, A', A")** Immunostaining of the Canton-S pupal retina with anti-Uch antibody, anti-Dlg, and Merge image, respectively. **(B, B', B")** Immunostaining of the Canton-S pupal retina with anti-Uch antibody, anti-Dlg, and Merge image, respectively. **(C, C', C")** Immunostaining of the Canton-S pupal retina with anti-Uch antibody, anti-Chp, and Merge image, respectively. Scale bar: 10 μm.

The rough eye included several atypical morphological features such as altered ommatidial shape, clustering of the ommatidia, disorganized ommatidial arrangement, and loss of mechano-sensory bristles. The phenotypic score, which quantifies the degree of ommatidial disorganization, was significantly higher in *Uch* knockdown eyes compared with the control group (Fig 1F). These findings strongly suggest a functional link between Uch loss and disrupted eye development in *D. melanogaster*.

### Distribution of Uch in the 42-h APF retina

To determine the location of Uch in the retinae at 42 h after puparium formation (APF), immunofluorescence staining was performed using an anti-Uch (Fig 2A–C) in combination with either anti-Chp (which binds specifically to chaoptin, a glyco-protein localized to the photoreceptor cell membrane) (Fig 2C') or Dlg antibody (which binds specifically to Discs-large, a membrane-associated marker) (Fig 2A' and B') in wild-type Canton-S flies. The results showed that Uch was expressed in the ommatidia of 42-h APF retinae, with a stronger signal observed in the interommatidial cells and mechanosensory bristles (Fig 2A"–C"). Uch was also

detected around the membrane regions of the photoreceptor cells (Fig 2). Notably, the Uch signal was markedly reduced in eye cells of Uch RNAi flies, confirming the knockdown efficiency (Fig S2A and B).

### Uch knockdown in developing eye induced abnormality in all cell types of the 42-h APF retina

The observation of the rough eye phenotype prompted us to further investigate the apical architecture of the retina under Uch knockdown conditions by staining the 42-h pupal retina with anti-Discs-large antibody. The results revealed that ommatidia in Uch knockdown retina were not arranged in the typical hexagonal lattice (Fig 3B and B'), and ommatidial orientation was irregular (Fig 3C and C'). Furthermore, many cone cell clusters contained an abnormal number of cells, either fewer or more than the usual four (Fig 3D and D'), and primary pigment cells appeared distorted (Fig 3D and D'). Compared with control retinae (Fig 3A and E), the interommatidial cells exhibited aberrant morphology and orga-nization. Distinguishing secondary and tertiary pigment cells be-came difficult, as these cells appeared swollen and increased the

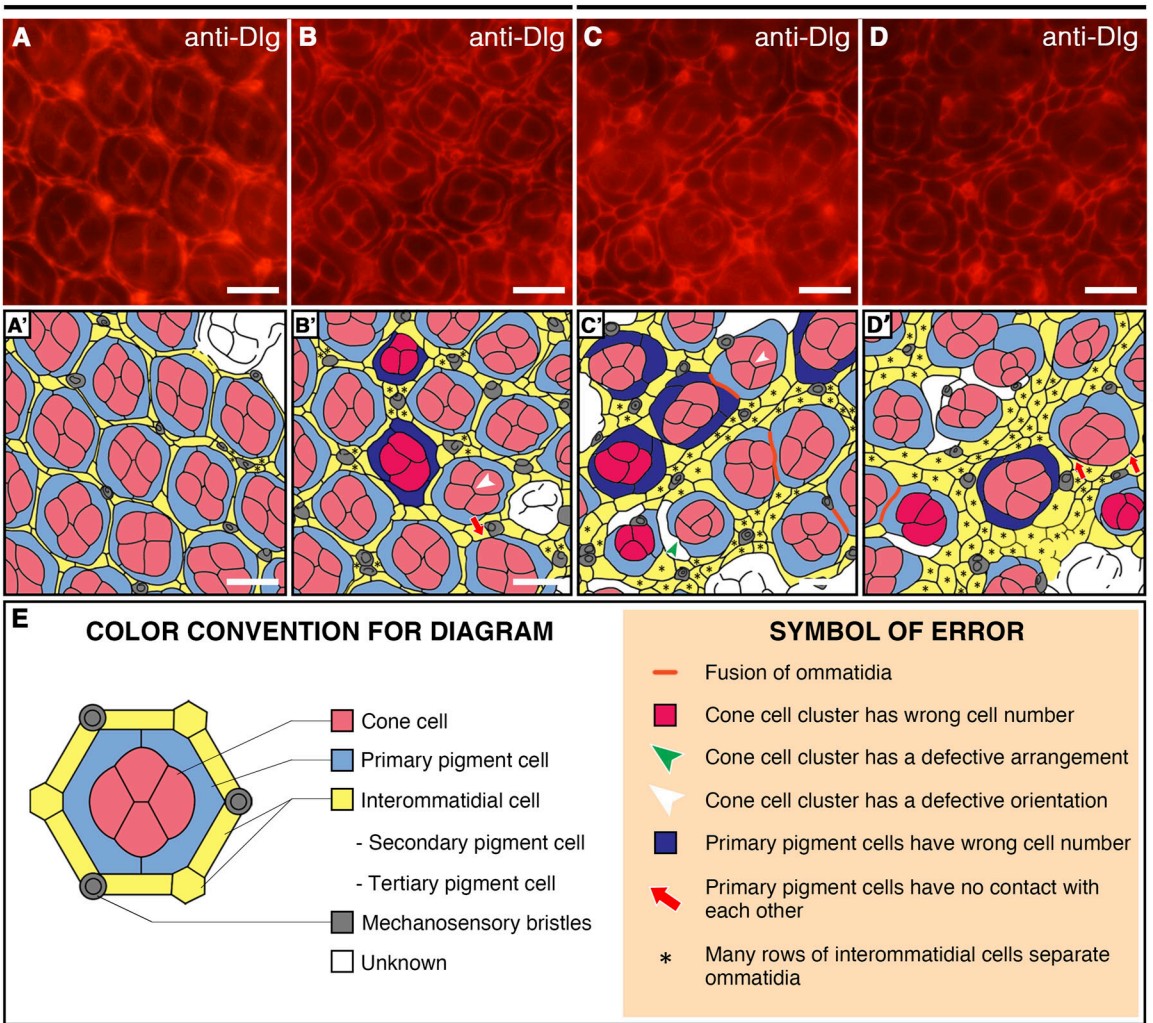

**Figure 3. Knockdown of *Uch* generated abnormalities in the apical pattern in the 42-h retinal pupae.**
**(A, B)** Control fly, GMR-GAL4>UAS-GFP-IR. **(C, D)** Uch knockdown fly, GMR-GAL4>UAS-Uch-IR. **(A, B, C, D, A', B', C', D')** Diagram was plotted and analyzed based on the corresponding immunostaining results in (A, B, C, D). Scale bar: 10 μm. **(A', B', C', D', E)** Color convention of the diagram (left) for the structure of ommatidium is shown in (A', B', C', D'), and the symbol for defects (right) is shown in (A', B', C', D').

spacing between adjacent ommatidia (Table 1 and Fig 3). These observations indicate that loss function of Uch disrupts ommatidial orientation and impairs the proper differentiation of pupal retinal cells, including cone cells, as well as primary, secondary, and tertiary pigment cells.

### Uch knockdown affected the development of cone cells and photoreceptor cells

The role of Uch in cone cell differentiation was investigated through immunofluorescence staining of a 42-h pupal retina using Cut antibody, a known marker for cone cells (23).

In Uch knockdown flies, cone cells appeared irregularly distributed and misoriented when compared to the organized pattern observed in a control line (Fig 4A and A'). In addition,

numerous cell clusters exhibited abnormal cone cell numbers (Fig 4B and B'). Quantitative analysis revealed that the proportion of cone cell clusters with aberrant cell number was significantly higher in Uch knockdown eyes than in control (Welch's *t* test, *P* < 0.0001) (Fig 4C).

In addition, abnormalities in photoreceptor cells of the Uch knockdown flies were examined. The results revealed that photoreceptor cells exhibited disorganized arrangement, irregular distribution, clusters with abnormal cell numbers, and instances where photoreceptor cell clusters appeared fused together (Fig 5C, C', D, and D') compared with the orderly pattern seen in control flies (Fig 5A, A', B, and B'). Similar defects in both cone and photoreceptor cells were also observed when **Uch** was knocked down using a different RNAi line, KK#103614 (Figs S3A and B and S4A and B).

**Table 1.  Uch knockdown retinal abnormalities.**

| Subject | Abnormal features |
|---|---|
| Ommatidium | - Messy distribution |
| | - Varying sizes |
| | - Irregular orientation |
| Cone cell cluster | - Distorted shape |
| | - Wrong cell number |
| | - Cell arrangement defect |
| | - Junction orientation defect |
| | - Cluster orientation defect |
| Primary cell | - Distorted shape |
| | - Wrong cell number |
| Interommatidial cell | - Irregular shape |
| | - Multiple rows between adjacent ommatidia |
| | - Disappeared and caused ommatidium fusion |

### Co-knockdown of rhomboid-1 and Uch rescued the rough eye phenotype

The differentiation of eye cells is tightly regulated by Notch, Sevenless, and EGFR signaling pathways. Among these, EGFR signaling has been reported as crucial for photoreceptor cell specialization from R1 to R7, excluding R8, whereas Notch and Sevenless specifically regulate R7 but not R1 to R6. Similar to Notch, EGFR signaling also plays a role in cone and pigment cell differentiation (11, 19). Given this, we investigated the genetic interactions between loss of Uch function and the EGFR pathway. Interestingly, among several EGFR-related genetic factors

tested, co-knockdown of Uch and rhomboid-1—which encode a protease responsible for cleaving the Spitz ligand of EGFR—completely rescued the rough eye phenotype caused by Uch knockdown (Fig 6A–K). This suggests that loss of rhomboid-1 function can mitigate the effects of Uch reduction. To confirm that the rescue effect was specifically due to the reduction of rhomboid-1 rather than off-target effects, different RNAi lines targeting rhomboid-1 were individually tested and yielded consistent results, fully rescuing the rough eye phenotype (Fig S5A–F). These findings indicate a crosstalk between the Uch protein and the EGFR signaling pathway mediated through rhomboid-1. To further validate the link between Uch and rhomboid-1, RT–qPCR analysis was performed, revealing that *rhomboid-1* transcription was up-regulated in the eyes of Uch knockdown flies (Fig S6).

### Co-knockdown of *rhomboid-1* rescued retinal apical abnormalities induced by the knockdown of *Uch*

To further investigate whether the reduction of rhomboid-1 can alleviate the retinal structural defects caused by Uch knockdown, we labeled 42-h pupal retinae with anti-Chp, Dlg, and Cut antibodies. Remarkably, our results showed that *rhomboid-1* knockdown completely suppressed the abnormal phenotypes induced by *Uch* knockdown. These included defects in photoreceptor cells (Fig 7A–E), disorganization in the arrangement, distribution, and orientation of apical retinal cells, adhesion of cell clusters (Fig 8A–D), and abnormal cone cell numbers within clusters (Fig 9A–E). Similar defects in photoreceptor cells, ommatidial arrangement, and cone cells were also observed with other Uch RNAi lines (Figs S3A and B', S4A and B', and S7A and B). Collectively, these

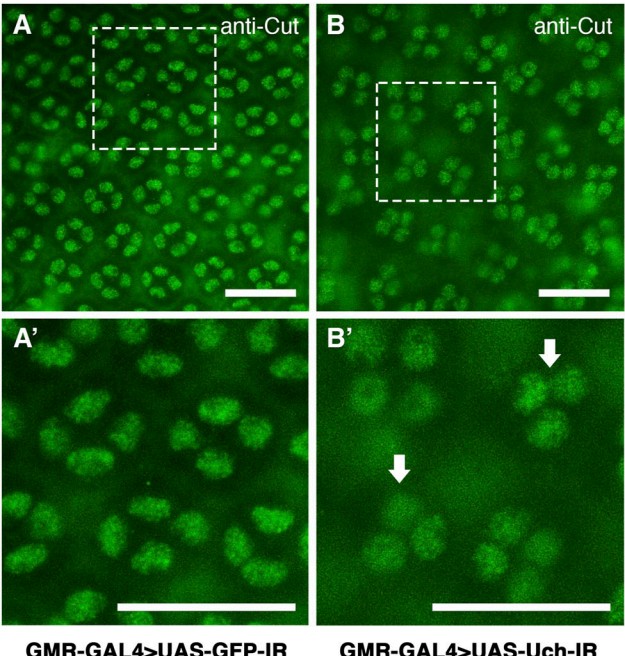

GMR-GAL4>UAS-GFP-IR          GMR-GAL4>UAS-Uch-IR

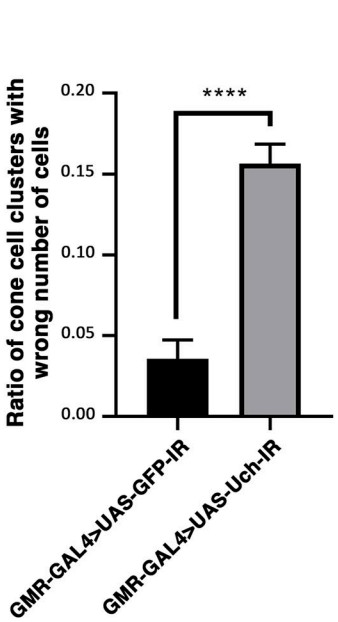

**Figure 4.  Uch knockdown increased the ratio of ommatidia carrying an abnormal number of cone cells.**
**(A, B)** Immunofluorescence staining of the 42-h pupal retina with the anti-Cut antibody of a GMR-GAL4>UAS-GFP-IR line (A) and a GMR-GAL4>UAS-Uch-IR line (B). **(A, A', B, B')** Magnification of the framed section in (A, A', B, B'). The white arrows indicate clusters of cone cells with abnormal numbers. **(C)** Graph showing the rate of abnormal clusters on the 42-h pupal retina of GMR-GAL4>UAS-GFP-IR (0.036; SD = 0.012; n = 10) and GMR-GAL4>UAS-Uch-IR (0.156; SD = 0.012; n = 5), the error bar represents the SD. Statistical analysis was carried out by a *t* test, ****$P$ < 0.0001. Scale bar: 20 $\mu$m.

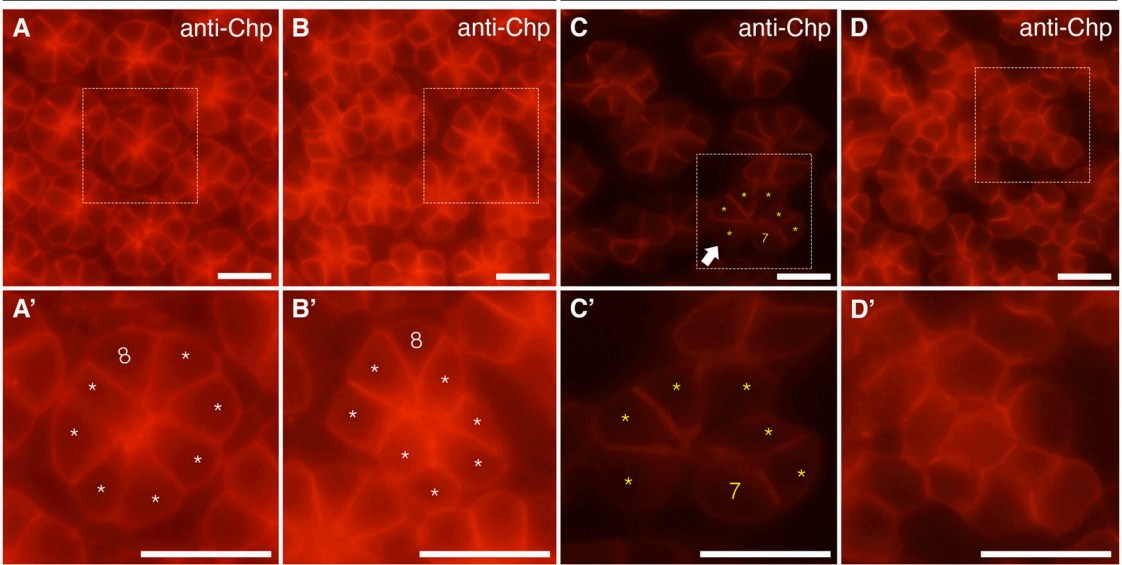

**Figure 5. Effect of Uch knockdown on photoreceptor cells of the 42-h pupal retina.**
**(A, B, C, D)** Result of retinal immunofluorescence staining with anti-Chp antibody. White arrows point to clusters of seven cells. **(A', B', C', D')** Enlarged image of the framed section in the corresponding figures. The asterisks and numbers aid in annotating the cell number in the cluster. Scale bar: 10 μm.

findings strongly indicate that loss of Uch function leads to significant abnormalities during fly eye development.

## Discussion

Ubiquitin carboxyl-terminal hydrolase L1 (UCH-L1) is a protein implicated in cancer, metabolism, and neurodegenerative diseases. Previous studies have demonstrated that knockdown of Uch induces a rough eye phenotype in *D. melanogaster*. Loss of Uch also results in abnormalities in the eye, wing, and thorax, as well as locomotor dysfunction in *Drosophila* (24). However, the precise mechanisms by which UCH-L1 regulates tissue development remain largely unclear. In this study, we provide evidence supporting a critical role of Uch in the development of the *Drosophila* eye.

Firstly, our data reconfirmed that the reduction of Uch causes a rough eye phenotype in flies. This phenotype was induced by knocking down Uch expression using two independent RNAi lines, and was rescued when Uch protein levels were restored by the UAS-Uch transgene. These results strongly demonstrate that loss of Uch function leads to the rough eye phenotype, consistent with previous reports (24). Together with studies on *Uch* overexpression (9), our findings suggest that maintaining a balanced level of Uch is essential for proper eye development in *Drosophila*.

The *Drosophila* ommatidium consists of eight photoreceptor cells, four cone cells, two primary pigment cells, six secondary pigment cells, three tertiary pigment cells, and three bristles (21, 25). Notably, this study revealed that loss of Uch led to abnormalities in retinal apical architecture, including disorganized distribution, variable cell sizes, and irregular orientation within the ommatidia. Flies with *Uch* knockdown in the eye imaginal disk

exhibited distorted cone and primary pigment cells, accompanied by incorrect cell numbers. Similar defects were consistently observed using different Uch RNAi lines (Figs S3, S4, and S7). These findings strongly indicate that Uch plays a critical role in *Drosophila* eye development.

In a previous report, Patterson et al discussed the roles of the proteostasis network in modulating developmental signaling (26), highlighting that the ubiquitin–proteasome system (UPS) plays multiple roles in cell growth and differentiation and is involved in *Drosophila* eye development. As noted above, UCH-L1 belongs to the DUB family, which may also contribute to eye development. Furthermore, previous studies demonstrated that Fat facets (Faf), a DUB, and its substrate Liquid facets (Lqf) participate in Delta-Notch signaling, a key pathway regulating eye differentiation (27). Notably, Lqf undergoes ubiquitination during eye development.

Loss of Faf function leads to impaired deubiquitinase activity, resulting in an excess number of photoreceptor cells within the ommatidium (28). The Notch and EGFR signaling pathways regulate eye cell specialization, ommatidial arrangement, and apoptosis to maintain proper cell numbers in each ommatidium. Aberrant ubiquitination of certain components in these pathways can cause defects in *Drosophila* eye development (26). Among several signaling pathways, including Notch and Sevenless, EGFR signaling plays a crucial role in eye differentiation, as it governs the specialization of all eye cell types except R8 (18, 22). In addition, EGFR signaling controls eye cell arrangement, distribution, and junction orientation (29). In this study, our data demonstrated that the reduction of *Uch* expression in the eye imaginal disk led to disorganized ommatidial distribution and fusion, defects in cone cell junction orientation, and abnormal photoreceptor cell arrangement. These results reveal a potential link between EGFR signaling and Uch in regulating eye development. Because loss of function of

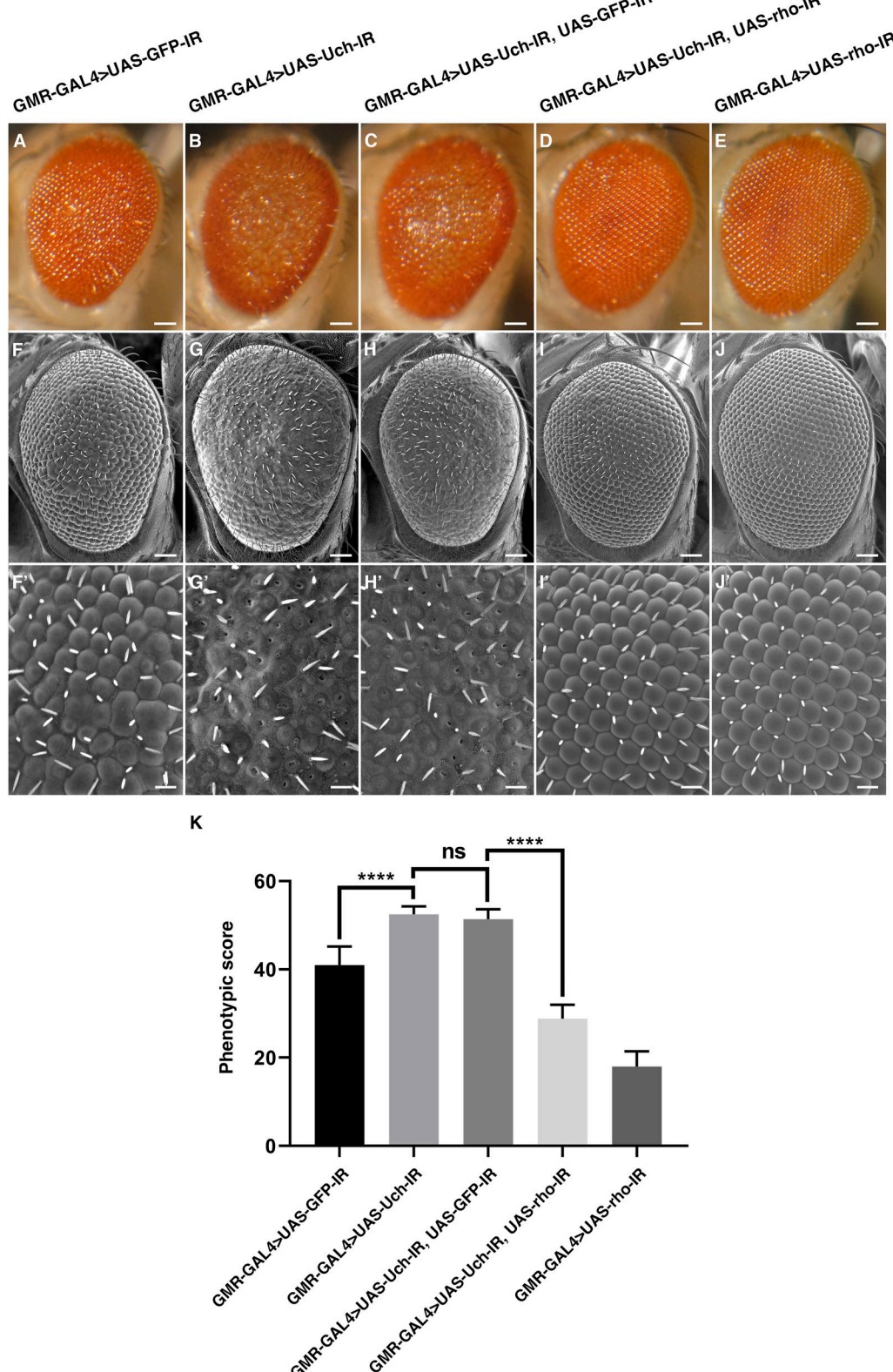

**Figure 6. Genetic interaction between Uch and rhomboid-1.**
**(A, B, C, D, E)** Stereoscopic images of adult fly eyes; scale bar: 50 $\mu$m. **(F, G, H, I, J)** Scanning electron microscope (SEM) images of the adult fly eyes; scale bar: 50 $\mu$m. **(F', G', H', I', J')** Magnification SEM image in the central region of the fly eye; scale bar: 14.2 $\mu$m. **(A, B, C, D, E, F, G, H, I, J, F', G', H', I', J')** GMR-GAL4>UAS-GFP-IR; (B, G, G') GMR-GAL4>UAS-Uch-IR; (C, H, H') GMR-GAL4>UAS-Uch-IR, UAS-GFP-IR; (D, I, I') GMR-GAL4>UAS-Uch-IR, UAS-rho-IR; and (E, J, J') GMR-GAL4>UAS-rho-IR. **(K)** Graph showing

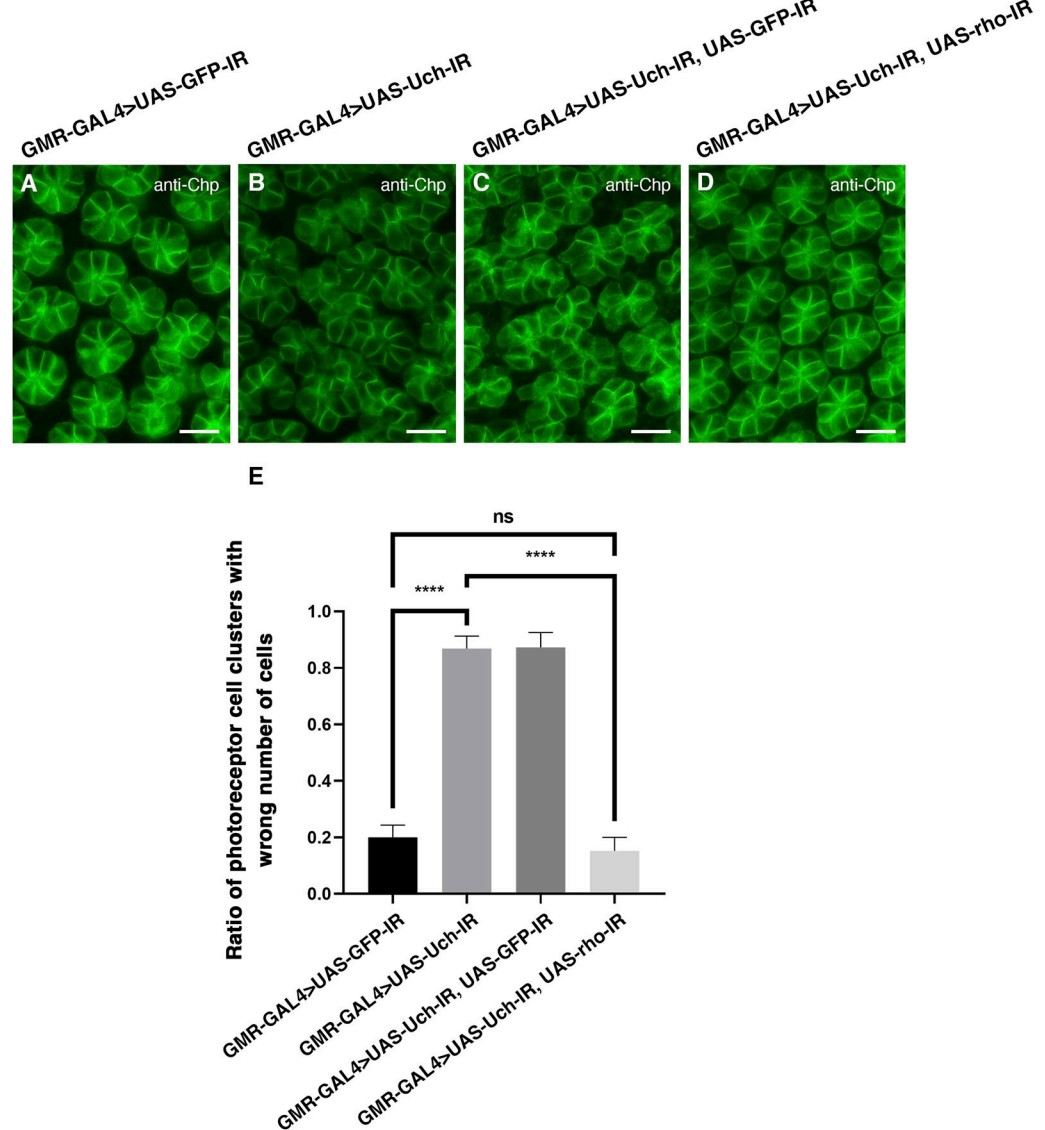

**Figure 7. Co-knockdown of *rhomboid-1* and *Uch* resolved the abnormal phenotype in the photoreceptor cells caused by Uch knockdown.**
**(A, B, C, D)** Result of retinal immunofluorescence staining with anti-Chp antibody of a GMR-GAL4>UAS-GFP-IR line (A), GMR-GAL4>UAS-Uch-IR line (B), GMR-GAL4>UAS-Uch-IR, UAS-GFP-IR line (C), and GMR-GAL4>UAS-Uch-IR, UAS-rho-IR line (D). **(E)** Ratio of photoreceptor cell clusters with wrong cell number (scale bar: 10 $\mu$m) of GMR-GAL4>UAS-GFP-IR (0.1996; SD = 0.043; n = 4), GMR-GAL4>UAS-Uch-IR (0.8686; SD = 0.043; n = 6), GMR-GAL4>UAS-Uch-IR, UAS-GFP-IR (0.8727; SD = 0.052; n = 7), and GMR-GAL4>UAS-Uch-IR, UAS-rho-IR (0.1518; SD = 0.048; n = 6); the error bar represents the SD. Statistical analysis was carried out by an ordinary one-way ANOVA, $P < 0.0001$, followed by Tukey's multiple comparisons post-test, ****$P < 0.0001$.

Ru causes a severe eye phenotype, the effects of co-knockdown of Uch and Ru could not be clearly distinguished (Fig S8C'). Remarkably, reduction of rhomboid, a protease that cleaves Spitz—the ligand of EGFR (30)—partially rescued the defects caused by Uch knockdown in ommatidia. The rough eye phenotype observed in Uch knockdown flies was completely rescued. Furthermore, abnormalities in cone cell clusters, ommatidial arrangement, and photoreceptor cells were also ameliorated. These findings suggest that Uch may function as a negative regulator of the EGFR signaling pathway during eye development. Knockdown

phenotypic scores analyzed by Flynotyper software from SEM images of eyes of a GMR-GAL4>UAS-GFP-IR line (40.97; SD = 4.197; n = 31); GMR-GAL4>UAS-Uch-IR line (52.46; SD = 1.79; n = 36); GMR-GAL4>UAS-Uch-IR, UAS-GFP-IR line (51.37; SD = 2,248; n = 35); GMR-GAL4>UAS-Uch-IR, UAS-rho-IR line (28.81; SD = 3.186; n = 39); and GMR-GAL4>UAS-rho-IR line (17.97; SD = 3.462; n = 35). Statistical analysis was carried out by Welch's ANOVA, $P < 0.0001$, followed by Games–Howell's multiple comparisons post-test, ns: $P = 0.1731$, ****$P < 0.0001$.

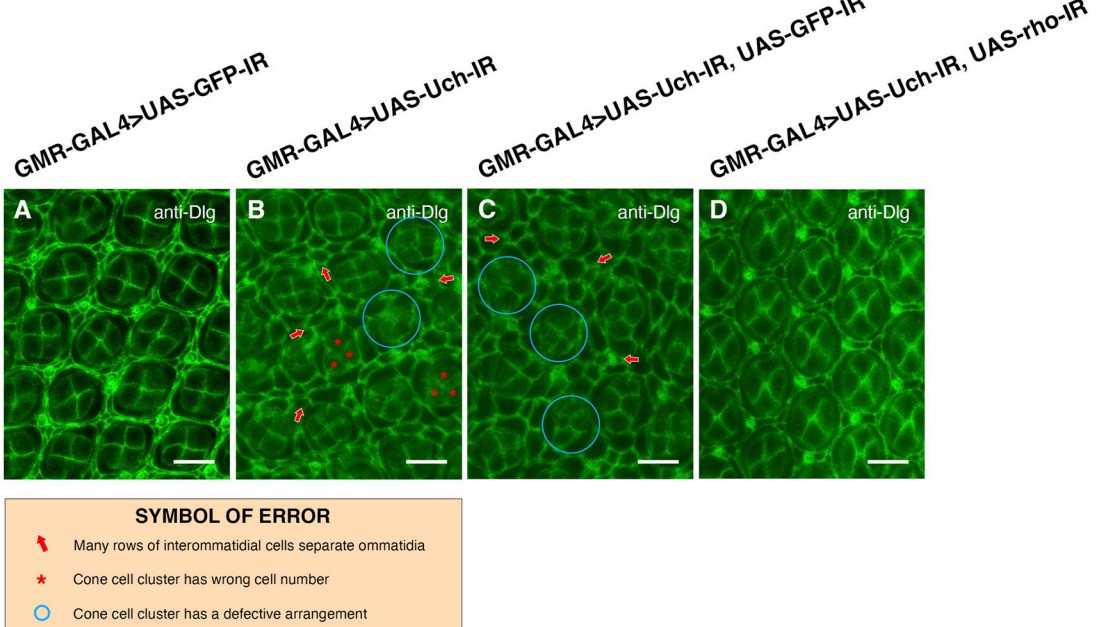

**Figure 8. Co-knockdown of *rhomboid-1* and *Uch* resolved the abnormal ommatidial arrangement caused by Uch knockdown.**
**(A, B, C, D)** Result of retinal immunofluorescence staining with anti-Dlg antibody of a GMR-GAL4>UAS-GFP-IR line (A), GMR-GAL4>UAS-Uch-IR line (B), GMR-GAL4>UAS-Uch-IR, UAS-GFP-IR line (C), and GMR-GAL4>UAS-Uch-IR, UAS-rho-IR line (D). Scale bar: 10 μm.

of Uch likely enhances EGFR signaling, resulting in severe eye defects. This enhancement is counteracted by reduction of rhomboid, as demonstrated in our study. Supporting this notion, we also observed partial rescue of the eye defects when the Uch knockdown phenotype was mitigated by the Spitz mutant, *spi^1* (Fig S8A, A′, B, and B′).

Uch is the *D. melanogaster* homolog of human UCH-L1. Although previous studies have indicated the importance of Uch in tissue development (24), its specific role in eye development remained unclear. The data presented in this study clearly demonstrate that Uch plays a crucial role in *Drosophila* eye development. Loss of Uch resulted in defects across multiple cell types within the ommatidium, including cone cells, photoreceptor cells, and pigment cells. These findings provide valuable insights into the function of Uch in fly eye development and establish a foundation for future research into the role of UCH-L1 in human biology.

## Materials and Methods

### Fly strains and maintenance

The *Uch* gene was knocked down by the GAL4-UAS system combined with the RNAi effect, in which the virgin female flies that owned the driver GAL4 under the control of the glass multiple reporter (GMR) promoter specific for the eye imaginal disk were crossed with UAS-RNAi transgenic male flies to give a single copy of RNAi background (heterologous). Only male offspring were used in experiments. All fly stocks were cultured on standard food carrying 5% (wt/vol) yeast extract, 5% (wt/vol) sucrose, 3% (wt/vol) milk powder, 0.8% (wt/vol) agar, 0.1% (wt/vol) sodium benzoate, and 0.5% (vol/vol) propionic acid at 25°C. All experimental lines were cultured on the same food at 28°C.

*Drosophila* fly strains were collected from Bloomington *Drosophila* Stock Center (BDSC), Vienna *Drosophila* Resource Center (VDRC), and Kyoto Stock Center (Kyoto): Canton-S (#64349; BDSC), GMR-GAL4 on the X chromosome (#16; Kyoto; a gift from Yamaguchi et al, Department of Applied Biology, Advanced Insect Research Promotion Center, Kyoto Institute of Technology) (31), UAS-GFP[RNAi] (#9330; BDSC), UAS-Uch[RNAi] (GD#26468 and KK#103614; VDRC), GMR-GAL4 (#16), UAS-rho-IR/TM3 (v51953; VDRC), UAS-rho-IR/CyO (v107502; VDRC), spi[1]/CyO (#1859; BDSC).

### Fly eye imaging using scanning electron microscope (SEM) and phenotypic analysis

Adult flies of 2 to 5 d old were anesthetized, mounted, and recorded by SEM Keyence VE-7800 (Kyoto Institute of Technology) in a vacuum condition. The images were processed by Flynotyper to generate phenotypic scores that reflect the disorderliness in ommatidial arrangement (32).

### Fly eye imaging using a stereomicroscope

Adult flies of 2 to 5 d old were anesthetized, mounted, and captured by a stereomicroscope Olympus SZX10 with a mounted camera Sony NEX-5R.

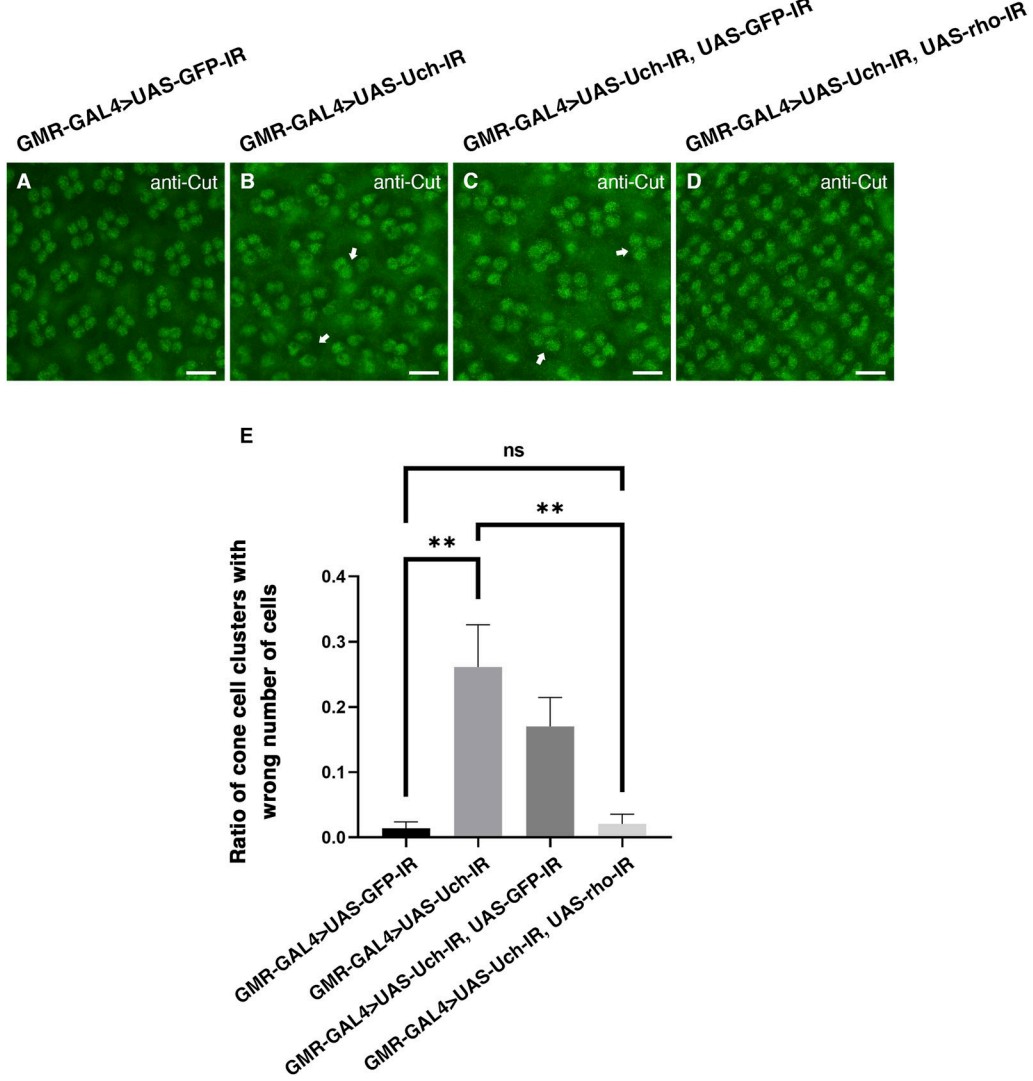

**Figure 9. Co-knockdown of *rhomboid-1* and *Uch* resolved the abnormal phenotype in the cone cell clusters caused by Uch knockdown.**
**(A, B, C, D)** Result of retinal immunofluorescence staining with anti-Cut antibody of a GMR-GAL4>UAS-GFP-IR line (A), GMR-GAL4>UAS-Uch-IR line (B), GMR-GAL4>UAS-Uch-IR, UAS-GFP-IR line (C), and GMR-GAL4>UAS-Uch-IR, UAS-rho-IR line (D). **(E)** Ratio of cone cell clusters with wrong cell number (scale bar: 10 μm) of GMR-GAL4>UAS-GFP-IR (0.0136; SD = 0.010; n = 7), GMR-GAL4>UAS-Uch-IR (0.2612; SD = 0.064; n = 5), GMR-GAL4>UAS-Uch-IR, UAS-GFP-IR (0.1702; SD = 0.044; n = 5), and GMR-GAL4>UAS-Uch-IR, UAS-rho-IR (0.0209; SD = 0.015; n = 7); the error bar represents the SD. Statistical analysis was carried out by Welch's ANOVA, followed by Games–Howell's multiple comparisons post-test, **$P$ = 0.0032.

## Immunofluorescence staining

The eye imaginal disk of the third larvae and retina at 42-h pupae were dissected in cold PBS 1X and fixed with 4% (wt/vol) or 4.6% (wt/vol) paraformaldehyde/PBS for 20 or 30 min at 25°C, respectively. After being washed three times with 0.3% Triton X-100/PBS for 20 min each time, the sample was blocked with 0.15% Triton X-100/PBS and 10% goat serum for 30 min at 25°C. The primary antibody for each experiment was added and incubated for 18–20 h at 4°C; the following antibodies were used: anti-chaoptin 1:100 (24B10-DSHB), anti-Discs-large 1:500 (4F3-DSHB), anti-Cut 1:500 (2B10-DSHB), anti-Uch 1:500 (HCMUS) (33). They were repeatedly washed five times with 0.3% Triton X-100/PBS and then incubated with 0.15% Triton X-100/PBS and 10% goat serum plus Alexa 488 or

Alexa 594 1:400 (Invitrogen) secondary antibody for 18–20 h at 4°C in sealed tubes to avoid light. After being washed with 0.3% Triton X-100/PBS five times, the pupal retina was dissected and mounted in VECTASHIELD Mounting Medium (Vector Laboratories). The fluorescence signal was recorded at a single plane of focus by ECLIPSE Ni-U (Nikon) and DS-Ri2 camera (Nikon) with an objective lens 40X and a numerical aperture 0.75.

## Quantitative RT–qPCR

For RT–qPCR analysis, 240 eye imaginal disks of the third late larvae (~4.5 d after laying eggs) were dissected per fly line. The total mRNA from larval eye disks was then extracted using TRIsure (Meridian) and phenol–chloroform methods. After that, the total mRNA was

reverse-transcribed to cDNA using PrimeScript RT Reagent. The mRNA and cDNA concentrations were measured by NanoDrop 1000 (Thermo Fisher Scientific). The primer set of F-R (5′-GAAC-TAATCGCCTCTCGCTATG; 5′-CAGTTTGCTGATGCTTCGATTC) was used. dRP49 was used as a reference gene of F-R (5′-AGATCGTGAA-GAAGCGCACC; 5′-CGATCCGTAACCGATGTTGG). Three replications of the RT–qPCR were conducted. The final result was analyzed by the $2^{-\Delta\Delta Cq}$ method.

### Statistical analysis

The statistical analysis and graphing were performed using GraphPad Prism 10.4.1 (GraphPad) software. The ratio of abnormal ommatidia (Fig 1) and the ratio of cone cell clusters (Fig 4) were analyzed by a *t* test because the data were unpaired and not restricted. Phenotypic scores (Fig 6) and the ratio of cone cell clusters with an incorrect number of cells (Fig 9) were analyzed by Welch's ANOVA, followed by Games–Howell's multiple comparisons post-test, because the five groups of data in the experiments were unpaired and unrestricted, and showed unequal dispersion. The ratio of photoreceptor cell clusters with an incorrect number of cells (Fig 7) was analyzed by ordinary one-way ANOVA ($P < 0.0001$), followed by Tukey's multiple comparisons post-test, because the groups of data values showed homogeneity of variances, indicating equal dispersion. The mRNA levels of *Uch* and *rhomboid-1* were analyzed by Welch's test, because the two groups of values showed heterogeneity of variances, which resulted in unequal dispersion. A *P*-value lower than 0.05 was considered statistically significant. Mean data are presented with the SD of independent samples per group.

## Data Availability

The data that support the findings of this study are available from the corresponding author, upon reasonable request.

## Supplementary Information

## Acknowledgements

The authors thank all members of the Gene Technology and Application Group, especially the Fly Biomedical Research Group, for their great support in this study. This research was funded by Vietnam National University, Ho Chi Minh City (VNU-HCM), under grant number 562-2024-18-09.

### Author Contributions

TTT Cao: data curation, formal analysis, and methodology.
TA Nguyen: data curation.
MHC Nguyen: data curation.
TTH Ngo Trang: project administration.
TTP Dang: conceptualization, formal analysis, supervision, funding acquisition, validation, investigation, methodology, project administration, and writing—original draft, review, and editing.

### Conflict of Interest Statement

The authors declare that they have no conflict of interest.

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
