## [Reviewer comments · Life Science Alliance]

Life Science Alliance

Critical roles of *Drosophila* Ubiquitin Carboxyl-terminal Hydrolase in eye development

Trang Thi Thuy Cao, Tuan Anh Nguyen, Minh Huy Cong Nguyen, Trang Thi Huyen Ngo and Thao Thi Phuong Dang

DOI: <https://doi.org/10.26508/lsa.202503246>

Corresponding author(s): Prof. Thao Thi Phuong Dang (Ho Chi Minh City University of Science)

Review Timeline:

Submission Date:	2025-02-04
Editorial Decision:	2025-03-11
Revision Received:	2025-05-14
Editorial Decision:	2025-07-01
Revision Received:	2025-08-01
Accepted:	2025-08-01

Scientific Editor: Sarita Hebbar

Transaction Report:

March 11, 2025

Re: Life Science Alliance manuscript #LSA-2025-03246-T

Thao Thi Phuong Dang
University of Science- VNU, Ho Chi Minh City

Dear Dr. Dang,

Thank you for submitting your manuscript entitled "Critical roles of Drosophila Ubiquitin Carboxyl-terminal Hydrolase in eye development" to Life Science Alliance. The manuscript was assessed by expert reviewers, whose comments are appended to this letter. We invite you to submit a revised manuscript addressing the Reviewer comments.

Thank you for this interesting contribution to Life Science Alliance. We are looking forward to receiving your revised manuscript.

Sincerely,

B. MANUSCRIPT ORGANIZATION AND FORMATTING:

Reviewer #1 (Comments to the Authors (Required)):

The paper reports a strong genetic dependency on EGF receptor signaling for phenotypic effects of depleting Ubiquitin Carboxy-terminal Hydrolase (Uch-L1) in the *Drosophila* eye. Uch-L1 is a deubiquitinating enzyme mainly expressed in the brain and implicated in human diseases including cancer, Parkinson's Disease, and Type-2 Diabetes. Here the authors show that depletion during *Drosophila* eye development results in morphological abnormalities, and that these depend to a large extent on Rhomboid, a rate-limiting enzyme in the production of the major EGF receptor ligand, strongly suggesting that the major developmental requirement for Uch-L1 is as a negative regulator of EGF receptor signaling.

The main results of the paper are quite clear and convincing. The results are potentially important in identifying an important pathway that depends on Uch-L1 and might be affected by Uch-L1 in human disease. The paper is limited in scope. Phenotypic analysis is largely based on gross anatomy, and there is little investigation of mechanism such as how Uch-L1 activity modulates EGFR signaling, or of the specific roles in each of the affected cell types. In principle, Uch-L1 might be affecting any of many steps in the processing and secretion of EGFR ligand, or in receptor activation or any step of signal transduction. The novelty of the results is undercut by the previous study (Thao et al 2012) that described a reduction in ERK signaling when Uch-L1 was over-expressed. It is certainly an advance to report that Uch-L1 loss of function affects the same pathway that was previously implicated downstream of Uch-L1 over-expression, but less impactful than if the connection between Uch-L1 and receptor tyrosine kinase signaling was being reported for the first time.

Specific comments.

Description of the Uch-L1 knockdown phenotype is quite clear.

Specificity of the immunostaining with anti-Uch-L1 should be confirmed using Uch-L1 knockdown.

Location of dUch-L1 protein in pupal retina is rather indistinct. One can see labeling of some cells in Figure 2B but I can't readily see pigment cells, or photoreceptor labeling. Better figures should be shown, perhaps with additional focal planes or additional markers to support the conclusions.

Suppression of the Uch-L1 knockdown phenotype by co-depletion of Rho is clear.

The authors mention that Rho was one of multiple signaling genes tested for interaction with Uch-L1 knock-down. They should list these other genes and their effects.

The paper would be more significant if the authors could narrow down where in the EGFR pathway is the potential Uch-L1 substrate, either by genetic epistasis analysis (as with Rhomboid) or perhaps by examining levels of EGFR pathway proteins.

Reviewer #2 (Comments to the Authors (Required)):

This manuscript investigates the role of the *Drosophila* Ubiquitin C-terminal hydrolase (Uch) in eye development. Reduction of Uch function by RNAi caused various defects in eye development, and these defects were rescued by simultaneous knockdown of Rhomboid, a protease that cleaves and releases Spitz, a ligand for EGFR. At present, this manuscript represents a limited advance in the field, essentially describing an outcome of two experiments, without any further mechanistic studies to explain the observed effects.

Major points:

A single RNAi line was used throughout the manuscript, but a second line is available and should be used in parallel to support the conclusions. Better yet, clones using a knockout allele should be investigated in addition to using RNAi.

Loss of Uch is likely to affect multiple pathways, and it is unclear why EGFR was focused on and other pathways (e.g. Notch) were not tested. Furthermore, to confirm that the EGFR pathway is involved, other components of the pathway should be tested in addition to Rhomboid. Also, at present there is no mechanistic explanation for a connection between Uch and Rho. At which level does reduction of Rho counteract loss of Uch?

Grammar and language should be checked and corrected throughout. Currently there are many awkwardly worded sentences in the manuscript.

Minor points:

Line 54: FlyBase nomenclature is Uch and should be used throughout.

Many panel legends inside the figures are too small, e.g. Fig. S1A, Fig. 1 and 6 (top legends), and others.

Fig. S1A: it is unclear what was used to visualize Uch - antibody staining? This information should be in the legend.

Line 98: "rescued" rather than "recused".

Fig. 1D,E are not referred to in the text.

Fig. 1E is too blurry, and this is a problem with some of the other light images of the fly eye. It is better to photograph fly eyes using an image stacking method, e.g. <https://www.biorxiv.org/content/10.1101/2024.01.26.577286v2>.

The quality of images in Fig. 2 is too low for making conclusions about the expression pattern of Uch.

Results in Fig. 7 should be quantified.

LETTER FOR SUBMISSION OF REVISED MANUSCRIPTS

Dr. Eric Sawey
Executive Editor

Life Science Alliance

Manuscript #LSA-2025-03246-T

Title: **Critical roles of Drosophila Ubiquitin Carboxyl-terminal Hydrolase in eye development**

Dear Dr. Eric Sawey,

Thanks for your email which pointed out some lines which help to improve our manuscript scientifically. Based on the comments from you and reviewers, we have revised our manuscript. Below are details of the revision.

Reviewer #1

1. Specificity of the immunostaining with anti-Uch-L1 should be confirmed using Uch-L1 knockdown.

We thank reviewer for the comment. In the revised manuscript, we added one more figure (Figure S2) to show the result of anti-Uch immunostaining in 42h retina of the Uch knocked down fly. The result clearly showed the reduction of anti-Uch signal in Uch loss function retina

2. Location of dUch-L1 protein in pupal retina is rather indistinct. One can see labeling of some cells in Figure 2B but I can't readily see pigment cells, or photoreceptor labeling. Better figures should be shown, perhaps with additional focal planes or additional markers to support the conclusions.

We thank reviewer for the comment. As suggestion from reviewer, we had changed the better figures to show the location of Uch in fly retina more clear

3. The authors mention that Rho was one of multiple signaling genes tested for interaction with Uch-L1 knock-down. They should list these other genes and their effects

We thank reviewer for the comment. In the previous version of manuscript, we presented data on interaction of one other EGFR pathway related gene (the Spitz) with knock down Uch. In this revised manuscript, we had added the results which obtained when we analysed the interaction between Uch loss function and other EGFR signaling related gene, the ru (Figure S7). However, due to the severe rough phenotype caused by single knock down ru, we could not have any conclusion on the possibility of interaction between the target genes. The Spitz mutant, spi¹, could partially rescue the rough eye phenotype

induced by Uch loss function. The loss function of Rho, as showed in the manuscript successfully rescued the effects of Uch loss function in fly eye.

4. The paper would be more significant if the authors could narrow down where in the EGFR pathway is the potential Uch-Li substrate, either by genetic epistasis analysis (as with Rhomboid) or perhaps by examining levels of EGFR pathway proteins.

Yes, we do strongly agree with reveiwer comment. Narrow down on the EGFR pathway to find out the target of Uch would be really interesting. In this revision verion, we added data which showed the up-regulation of Rhomboid gene expression in loss function Uch condition (Fig S8)

With the aims to address the effects of Uch loss function in eye development, we believe that data showed in this manuscript is strong enough. Further investigation the detail molecular mechanism will be performed later on. We hope the reviewer do agree with us in this point.

Reviewer #2

1. A single RNAi line was used throughout the manuscript, but a second line is available and should be used in parallel to support the conclusions. Better yet, clones using a knockout allele should be investigated in addition to using RNAi.

We thank reviewer for the comment. In the previous version of manuscript, we presented data of knock down Uch by using other Uch RNAi line (the KK #103614 – Figure S1B). The result clearly showed rough eye phenotype was caused by different line of Uch RNAi.

In the revised manuscript, we had added more data when we knoc down Uch by the second Uch RNAi KK #103614 line (Figure S3, S4, S5) The data clearly showed the defects on photoreceptor cells, cone cells and ommatidial orientation, arrangement.

2. Loss of Uch is likely to affect multiple pathways, and it is unclear why EGFR was focused on and other pathways (e.g. Notch) were not tested.

The eye cell differentiation process is tightly regulated by Notch, Sevenless and EGFR signaling. Among these, the EGFR signaling was reported as crucial signaling for photoreceptor cell specialization from R1 to R7, except R8 while Notch and Sevenless function on R7 but not R1 to R6. Similar to the Notch, the EGFR signaling also involved in cone and pigment cell differentiation. Besides that, previous studies reported that EGFR signaling also regulates eye cell arrangement, distribution, and junction orientation.

In this study, our data demonstrated that the reduction of Uch in the eye imaginal disc resulted in messy distribution and fusion of ommatidia, junction orientation defect of cone cells, and abnormal arrangement of photoreceptor cells.

Since all of the observed defects were realted to what reported as EGFR signaling regulation, those results given some hints of the possibility of a link between knockdown

Uch and EGFR signaling and took us uncovered if there is a link between EGFR signaling and Uch in controlling eye development. Other signalling such as Notch, Sevenless are also interesting for further study.

In the revised manuscript, we did rewrite some sentences to make the expansion more clear.

3. Furthermore, to confirm that the EGFR pathway is involved, other components of the pathway should be tested in addition to Rhomboid. Also, at present there is no mechanistic explanation for a connection between Uch and Rho. At which level does reduction of Rho counteract loss of Uch?

Yes, we thank for the comments from reviewer. We performed some experiments to analyse the interaction of Uch with other EGFR pathway components as showed in Figure S7. In which the Spitz mutant, *spi*¹, could partially rescue the rough eye phenotype induced by Uch loss function (Figure S7). However, due to the severe rough phenotype caused by single knock down *ru*, we could not have any conclusion on the possibility of interaction between these target genes. We also added a data of RT-qPCR which showed the upregulation of *rhomboid-1* mRNA when Uch was knocked down (Figure S8)

Uncover the mechanism of the connection between Uch and Rho could be very interesting. At the present due to the limitation of reagent we could not go further. We do understand that would be the limitation of the study. **However, setting the aims of this manuscript is addressing the effects of Uch loss function in eye development**, we do hope that data showed in this manuscript is strong enough and fit with the room of Life Science Alliance. Further investigation the detail molecular mechanism will be performed later on. We hope to have the sympathy from reviewer at this point.

4. Grammar and language should be checked and corrected throughout. Currently there are many awkwardly worded sentences in the manuscript.

Yes, we thank for the comments from reviewer. We are shy to have such of not good enough english writing. We did carefully check and revised throughout the manuscript. We did also ask for the native English editing to make the manuscript better.

5. Line 54: FlyBase nomenclature is Uch and should be used throughout. Many panel legends inside the figures are too small, e.g. Fig. S1A, Fig. 1 and 6 (top legends), and others.

Yes, we thank for the comments from reviewer. We made a changed throughout the manuscript. All of panel legends was edited. We have updated the new Figures to the system too.

6. Fig. S1A: it is unclear what was used to visualize Uch - antibody staining? This information should be in the legend.

Yes, we thank for the comments from reviewer. We made a change in the figure S1. In which, the anti-Uch was stated. We also described it clearly in figure legend

7. Line 98: "rescued" rather than "recused".

Fig. 1D,E are not referred to in the text. Fig. 1E is too blurry, and this is a problem with some of the other light images of the fly eye. It is better to photograph fly eyes using an image stacking method,

e.g. <https://www.biorxiv.org/content/10.1101/2024.01.26.577286v2>.

The quality of images in Fig. 2 is too low for making conclusions about the expression pattern of Uch.

Yes, we thank for the comments from reviewer. We had carefully read and made a changed for all of typos throughout the manuscript. We also mentioned a bout the Fig 1D, E in the maintext. Figure 1E and Figure 2 were replaced by higher quality figures.

8. Results in Fig. 7 should be quantified.

Yes, we thank for the comments. We added a quantification chart to show the ratio of abnormal cone cell clusters and abnormal photoreceptor cell clusters. Thereby, we sperated the data into 3 figures: Figure 7 – showed the resolve of the defects on photoreceptor cells which caused by Uch loss function; Figure 8 – showed the rescue of disorder arrangement which caused by Uch loss function; Figure 9- showed the rescued of defects in cone cell clusters.

We do hope our revision can satisfy you and meet the requirement for possible publication.

Once again, thank so much for your kind consideration on our manuscript.

Sincerely,

Dang Thi Phuong Thao, Ph.D

Faculty of Biology – Biotechnology

University of Science

Vietnam National University in Ho Chi Minh City

June 30, 2025

RE: Life Science Alliance Manuscript #LSA-2025-03246-TR

Prof. Thao Thi Phuong Dang
Ho Chi Minh City University of Science
227 Nguyen Van Cu, Dist 5
Ho Chi Minh, Ho Chi Minh 700000
Viet Nam

Dear Dr. Dang,

Thank you for submitting your revised manuscript entitled "Critical roles of Drosophila Ubiquitin Carboxyl-terminal Hydrolase in eye development". One of the reviewers has evaluated that the original concerns have been addressed in this revised manuscript, and also commented that the manuscript has limited new advance. We discussed this further with our academic editor. Our editorial decision is to publish your paper in Life Science Alliance pending final revisions necessary to meet our formatting guidelines.

- Please revise the figure legends for figures 4, S3, and S6, such that the figure panels are introduced in alphabetical order
- Please add callouts for Figures 2A-C; 3A-E; 4A; 5A-B; 6A-K; 7A-E; 9A-E; S2A-B; S3A-B; S4A-B; S5A-B; S6A-F and S7A to your main manuscript text
- Please include scale bar for all images provided in the figures.
- Please include a 'Data Availability' section. Please also specify if you are willing to provide source data or have done so in supplementary information.
- Please clearly indicate in the methods section if the RNAi-driven combinations had single copy of RNAi background (heterozygous)
- Please provide details for fluorescence microscopy imaging (objective used, plane selection etc) in the methods section
- In the methods section, please provide a section on statistical testing and state the name of the statistical test for the various comparisons, the number (n) of independent experiments underlying each data point (not replicate measures of one sample), and the P value for each test. In cases where n is small, a justification for the use of the statistical test employed has to be provided.
- Please provide a basic description of quantified values in various figures (n, mean, SEM). Specifically indicate the statistical significance (*) for figures 7 and 9
- Please do a thorough spell and grammar check on the entire manuscript.
- Supplementary figures should be uploaded only separately, and their legends should only appear in the manuscript file.
- Please add the X and Bluesky handles of your host institute/organization as well as your own or/and one of the authors in our system
- On the manuscript's title page, please provide the full name of each author, including middle names as initials, formatted as follows: First name, middle initial, Last name. This needs to match the system
- it is recommended to exclude figures from the manuscript text and upload them separately
- Please add your main, supplementary figure, and table legends to the main manuscript text after the references section;

A. FINAL FILES:

B. MANUSCRIPT ORGANIZATION AND FORMATTING:

Sincerely,

Sarita Hebbbar, PhD
Scientific Editor
Life Science Alliance
<http://www.lsjournal.org>

Reviewer #2 (Comments to the Authors (Required)):

My concerns have been largely addressed but I still think that the impact of this study is rather limited at this point.

August 1, 2025

RE: Life Science Alliance Manuscript #LSA-2025-03246-TRR

Prof. Thao Thi Phuong Dang
Ho Chi Minh City University of Science
227 Nguyen Van Cu, Dist 5
Ho Chi Minh, Ho Chi Minh 700000
Viet Nam

Dear Dr. Dang,

Thank you for submitting your Research Article entitled "Critical roles of Drosophila Ubiquitin Carboxyl-terminal Hydrolase in eye development". It is a pleasure to let you know that your manuscript is now accepted for publication in Life Science Alliance. Congratulations on this interesting work.

DISTRIBUTION OF MATERIALS:

Again, congratulations on a very nice paper. I hope you found the review process to be constructive and are pleased with how the manuscript was handled editorially. We look forward to future exciting submissions from your lab.

Sincerely,

Sarita Hebbar, PhD
Scientific Editor
Life Science Alliance
<http://www.lsajournal.org>